# Sperm-specific COX6B2 enhances oxidative phosphorylation, proliferation, and survival in human lung adenocarcinoma

Chun-Chun Cheng[1], Joshua Wooten[2], Zane A Gibbs[1], Kathleen McGlynn[1], Prashant Mishra[3], Angelique W Whitehurst[1]*

[1]Department of Pharmacology, Simmons Comprehensive Cancer Center, UT Southwestern Medical Center, Dallas, United States; [2]Nuventra, Durham, United States; [3]Children's Research Institute, UT Southwestern Medical Center, Dallas, United States

**Abstract** Cancer testis antigens (CTAs) are proteins whose expression is normally restricted to the testis but anomalously activated in human cancer. In sperm, a number of CTAs support energy generation, however, whether they contribute to tumor metabolism is not understood. We describe human COX6B2, a component of cytochrome c oxidase (complex IV). *COX6B2* is expressed in human lung adenocarcinoma (LUAD) and expression correlates with reduced survival time. COX6B2, but not its somatic isoform COX6B1, enhances activity of complex IV, increasing oxidative phosphorylation (OXPHOS) and NAD$^+$ generation. Consequently, COX6B2-expressing cancer cells display a proliferative advantage, particularly in low oxygen. Conversely, depletion of COX6B2 attenuates OXPHOS and collapses mitochondrial membrane potential leading to cell death or senescence. COX6B2 is both necessary and sufficient for growth of human tumor xenografts in mice. Our findings reveal a previously unappreciated, tumor-specific metabolic pathway hijacked from one of the most ATP-intensive processes in the animal kingdom: sperm motility.

*For correspondence:
angelique.whitehurst@
utsouthwestern.edu

## Introduction

Tumors frequently activate genes whose expression is otherwise restricted to testis; these genes are known collectively as cancer-testis antigens (CT-antigens, CTAs). The expression of these genes outside their native and immune privileged site has been the basis for immunotherapeutic approaches including vaccines and adoptive T-cell therapy (*Hunder et al., 2008*; *Gjerstorff et al., 2015*). Historically, functional knowledge of the contribution of these proteins, if any, to neoplastic behaviors has significantly lagged behind immune-targeting. In recent years, the functional characterization of CTAs has gained broader interest and these proteins have been implicated in tumor cell survival, TGF-β signaling, mitotic fidelity, polyadenylation, mRNA export, tumor suppressor silencing, and DNA damage repair (*Gibbs and Whitehurst, 2018*; *Viphakone et al., 2015*; *Nichols et al., 2018*). Multiple reports now indicate that anomalous expression of testis proteins can be engaged to support the tumorigenic program. The biased expression pattern of these proteins to tumors and testis may offer an extraordinarily broad therapeutic window if they can be targeted directly.

One unique aspect of mammalian sperm physiology is the tremendous energy demand required for motility while preserving the integrity of their precious DNA cargo within the hostile environment of the female reproductive tract. To meet this demand, sperm contain a number of tissue-specific protein isoforms for glycolysis and oxidative phosphorylation (OXPHOS) that mediate the increase in

ATP production. For example, lactate dehydrogenase C, LDHC, is a testis-specific isoform of the terminal enzyme in glycolysis that catalyzes the reduction of pyruvate to lactate essential for male fertility (*Odet et al., 2008*; *Krisfalusi et al., 2006*; *Li et al., 1989*). Similarly, COX6B2 is a testis-specific subunit of cytochrome c oxidase (complex IV) (*Hüttemann et al., 2003*). mRNA encoding of either *LDHC* or *COX6B2* is undetectable in normal tissue, but both are upregulated in a number of different tumor derived cell lines, classifying them as CTAs (*Maxfield et al., 2015*, (CTpedia (http://www.cta.lncc.br/index.php))). However, it is unclear whether these proteins support metabolic programs in tumor cells.

In a large-scale lossof-function analysis to annotate the contribution of individual CTAs to neoplastic behaviors, we found that COX6B2 is essential for survival of non-small cell lung cancer (NSCLC) cell lines (*Maxfield et al., 2015*). COX6B2 is a nuclear encoded, sperm-specific component of complex IV (*Hüttemann et al., 2003*). By transferring electrons from reduced cytochrome c to $O_2$, complex IV is the rate-limiting step for ATP production by the electron transport chain (ETC). Thirteen subunits make up complex IV: three are mitochondrial encoded and ten are derived from nuclear DNA. Six of the 10 nuclear encoded subunits have tissue-specific isoforms that permit regulation of this complex in response to environmental cues (e.g. pH, hormones, metals, ATP/ADP ratio etc.) (*Kadenbach and Hüttemann, 2015*). The somatic isoform of COX6B2 is COX6B1. These two proteins share 58% amino acid identity and ~80% similarity. Based on structural information, it is apparent that COX6B1/2 are the only complex IV subunits that are not membrane bound. Instead, their localization is confined to the intermembrane space where cytochrome c associates. It is inferred that COX6B1/B2 participate in dimerization of complex IV and cytochrome c association (*Sampson and Alleyne, 2001*; *Tsukihara et al., 1996*). While limited, studies on COX6B1 indicate that it is essential for complex IV activity. In particular, human mutations in COX6B1 (R20C or R20H) abrogate complex IV assembly and lead to severe cardiac defects (*Abdulhag et al., 2015*). In addition, biochemical analysis indicates that removal of COX6B1 from assembled complexes enhances complex IV activity, implying a negative regulatory role for COX6B1 (*Weishaupt and Kadenbach, 1992*). To-date no reports detail the function of COX6B2 nor indicate the physiological relevance of this sperm-specific subunit to fertility. Furthermore, there are no reports describing mechanism(s) that regulate *COX6B1* or *COX6B2* expression. Overall, compared to the enormous data on complex IV in general, the COX6B proteins have been understudied.

We have undertaken a detailed investigation into the mechanism of action of COX6B2 when this protein is aberrantly expressed in NSCLC. Here, we report that COX6B2 enhances mitochondrial oxidative phosphorylation (OXPHOS) in tumor cells. This activity accelerates proliferation in vitro and in vivo. In contrast, silencing of COX6B2 attenuates OXPHOS, reduces tumor cell viability and dramatically decreases growth in vivo. Importantly, we find that hypoxia enhances *COX6B2* expression, which confers a selective advantage for proliferation under low oxygen. Indeed, elevated COX6B2 mRNA correlates with reduced survival of LUAD patients. Cumulatively, this study demonstrates the remarkable capacity of tumor cells to integrate primordial gene products into their regulatory environment as a means of promoting unrestrained proliferation. In turn, COX6B2 becomes a liability that may be exploited for tumor selective targeting of OXPHOS.

## Results

### COX6B2 is expressed in human lung adenocarcinoma (LUAD) tumors and correlates with poor survival

Using a large-scale functional genomics approaches, we previously found that depletion of COX6B2 activates cleaved caspase 3/7 in cell lines derived from breast, melanoma and NSCLC, with the most potent activation in H1299 NSCLC cells (*Figure 1—figure supplement 1A*; *Maxfield et al., 2015*). Based on these findings, we examined the relationship between *COX6B2* expression and patient outcome in human NSCLC. Strikingly, elevated expression of *COX6B2* is associated with significantly shorter overall survival (OS) time (p=$5.3\times10^{-6}$, log rank test; hazard ratio (HR): 1.46; 95% confidence interval (CI): 1.24–1.73) (*Figure 1A*). In addition, *COX6B2* expression positively correlates with time to first progression (FP) (p=$4.1\times10^{-4}$, log rank test; HR: 1.63; 95% CI: 1.24–2.14) (*Figure 1A*). Separation of the two major histological subtypes of NSCLC revealed a strong correlation for both outcomes in LUAD (OS: p=$1.6\times10^{-4}$, log rank test; HR: 1.59; 95% CI: 1.25–2.03; FP: p=$9.4\times10^{-5}$, log

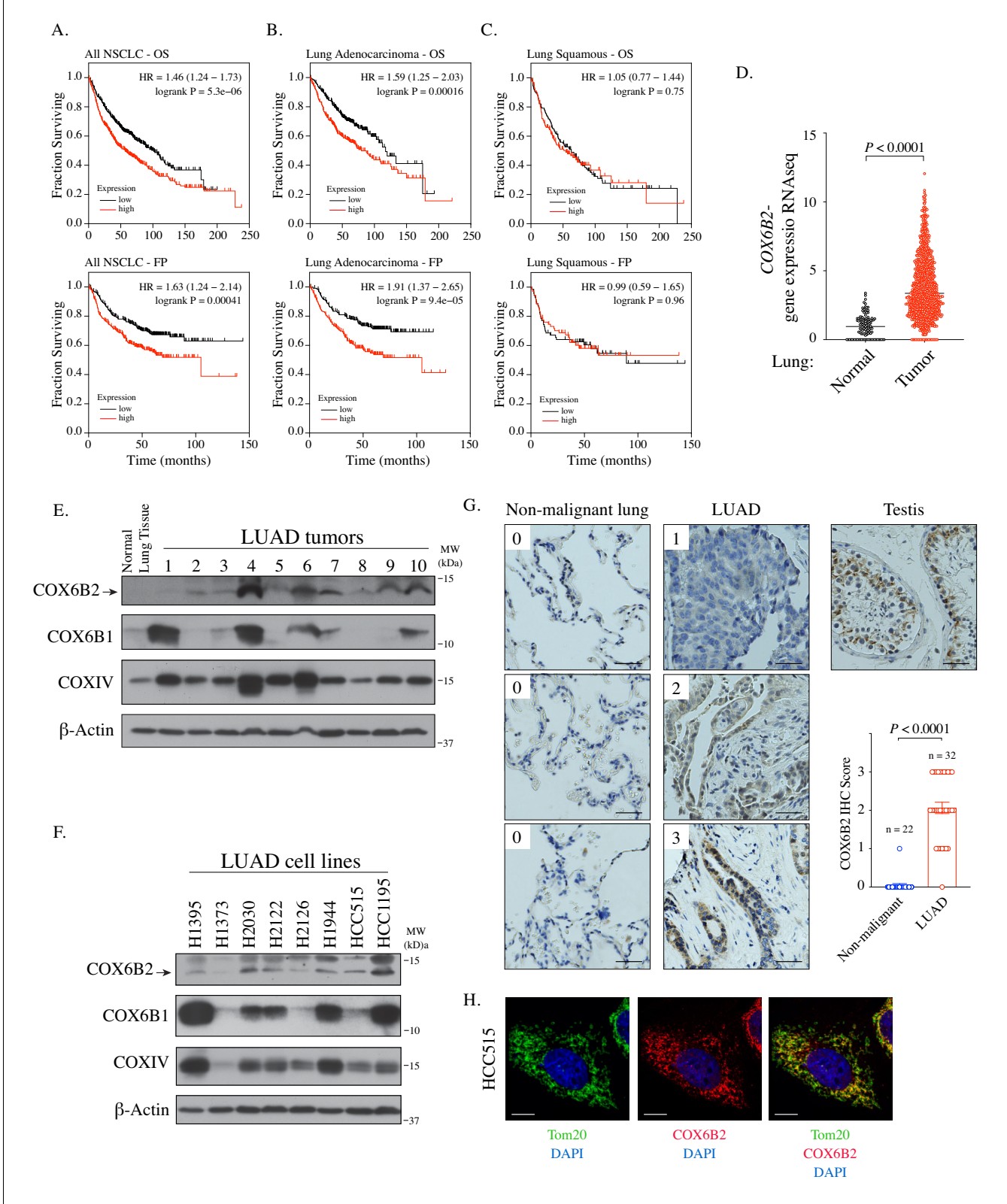

**Figure 1.** *COX6B2* mRNA expression correlates with poor outcome in LUAD. (A–C) Kaplan-Meier curves for OS and FP in NSCLC (A) LUAD (B) and LUSC patients (C). (D) *COX6B2* mRNA expression (RNA-seq RSEM, log2(norm count +1)) from TCGA Lung Cancer dataset. Bars represent median (normal: n = 110; tumor: n = 1017). *P* values calculated by Mann-Whitney test. (E–F) Whole tissue homogenates of LUAD tumors (E) and LUAD cell line lysates (F) were immunoblotted with indicated antibodies. Molecular weight (MW) markers are indicated. (G) IHC staining of non-malignant testis (a

*Figure 1 continued on next page*

*Figure 1 continued*

positive control), non-malignant lung (adjacent normal) and LUAD tissues. Scores ranged from 0 to 3. Scale bar, 50 µm. Bars represent mean ± SEM. p-Values calculated by Mann-Whitney test. (**H**) Representative confocal images of endogenous COX6B2 in HCC515 cells. Tom20 is used as a mitochondrial marker. Images were shown as Z-stack maximum projection from 0.3-µm-thick image. Scale bar, 10 µm.

The online version of this article includes the following figure supplement(s) for figure 1:

**Figure supplement 1.** COX6B2 expression in LUAD.

rank test; HR: 1.91; 95% CI: 1.37–2.65) (*Figure 1B*). However, this correlation is not observed in lung squamous carcinoma (LUSC) (OS: p=0.75, log rank test; HR: 1.05; 95% CI: 0.77–1.44; FP: p=0.96, log rank test; HR: 0.99; 95% CI: 0.59–1.65) (*Figure 1C*). Analysis of TCGA data sets also revealed a significant upregulation of *COX6B2* mRNA in LUAD tumors as compared to normal tissues (*Figure 1D*). We next immunoblotted for COX6B2 protein in a panel of human LUAD tumors and normal lung tissue using an antibody that detects COX6B2 (*Figure 1E*; *Figure 1—figure supplement 1B*). We found that COX6B2 protein expression is low or undetectable in normal lung tissue, but present in all tumor-derived tissues irrespective of stage and grade (*Figure 1E*; *Figure 1—figure supplement 1C*). A similar pattern is also observed in LUAD-derived cell lines (*Figure 1F*). Furthermore, immunohistochemical staining of COX6B2 in a panel of human non-malignant (n = 22) and LUAD tissues (n = 32) indicated accumulation of COX6B2 in tumor, but not non-malignant tissues (*Figure 1G*). This expression of COX6B2 in tumors does not have an apparent correlation with the presence or absence of COX6B1, the somatic isoform of COX6B2. We did observe a moderate correlation with the complex IV subunit, COXIV, suggesting COX6B2 accumulation coincides with increased mitochondria in cancer cells (*Figure 1—figure supplement 1D*). Furthermore, depletion or overexpression of either COX6B1 or COX6B2 does not impact protein accumulation of the corresponding isoform (*Figure 1—figure supplement 1B*). These findings indicate that while the isoforms are closely related in sequence their regulation may be through independent mechanisms. Immunofluorescence of endogenous COX6B2 in intact LUAD cells revealed specific localization to the mitochondria as judged by overlap with Tom20, a mitochondrial resident protein (*Figure 1H*). Thus, COX6B2 is frequently expressed in LUAD, correlates with poor survival and when ectopically expressed can localize to its native mitochondrial location.

## COX6B2, but not COX6B1, enhances oxidative phosphorylation in LUAD

To investigate whether COX6B2 supports OXPHOS when ectopically expressed in tumors, we generated a LUAD cell line that overexpresses COX6B2 protein ~3 fold (HCC515-COX6B2-V5) (*Figure 2A*). This overexpression is not associated with an increase in mitochondrial DNA or total mitochondria (*Figure 2B–C*). We then measured the oxygen consumption rate (OCR) at baseline and following exposure to electron transport chain perturbagens. This analysis revealed a significant enhancement in basal, ATP-linked, maximal and reserve OCR in COX6B2-V5 expressing cells as compared to control (*Figure 2D*). A similar trend is observed in an independent LUAD cell line, H2122, expressing COX6B2-V5 (*Figure 2E–F*). The OCR increase is associated with an elevation in total ATP in both cell lines (*Figure 2G*). In addition, we observed elevated NAD$^+$/NADH in COX6B2-V5 cells, indicative of an overall increase in electron transport activity (*Figure 2H*). We did not observe a significant or reproducible alteration in extracellular acidification rate (ECAR) in either setting (*Figure 2I*). We also studied whether COX6B2's somatic isoform, COX6B1, was sufficient to enhance OCR (*Figure 2J*). COX6B1-V5 expression appears to have no effect or slightly decreases OCR for each parameter (*Figure 2K*), consistent with prior biochemical studies where COX6B1 removal enhances the activity of complex IV (*Weishaupt and Kadenbach, 1992*).

## COX6B2 enhances complex IV activity and is incorporated into mitochondrial supercomplexes

To further characterize the impact of COX6B2 on complex IV, we directly assessed complex IV activity using a technique to pharmacologically isolate complex IV in permeabilized COX6B1-V5 or COX6B2-V5 cells (*Salabei et al., 2014*; *Divakaruni et al., 2014*; *Figure 3A*). Here, we find that COX6B2-V5 overexpression confers a significant increase (~25%) in complex IV activity in both LUAD

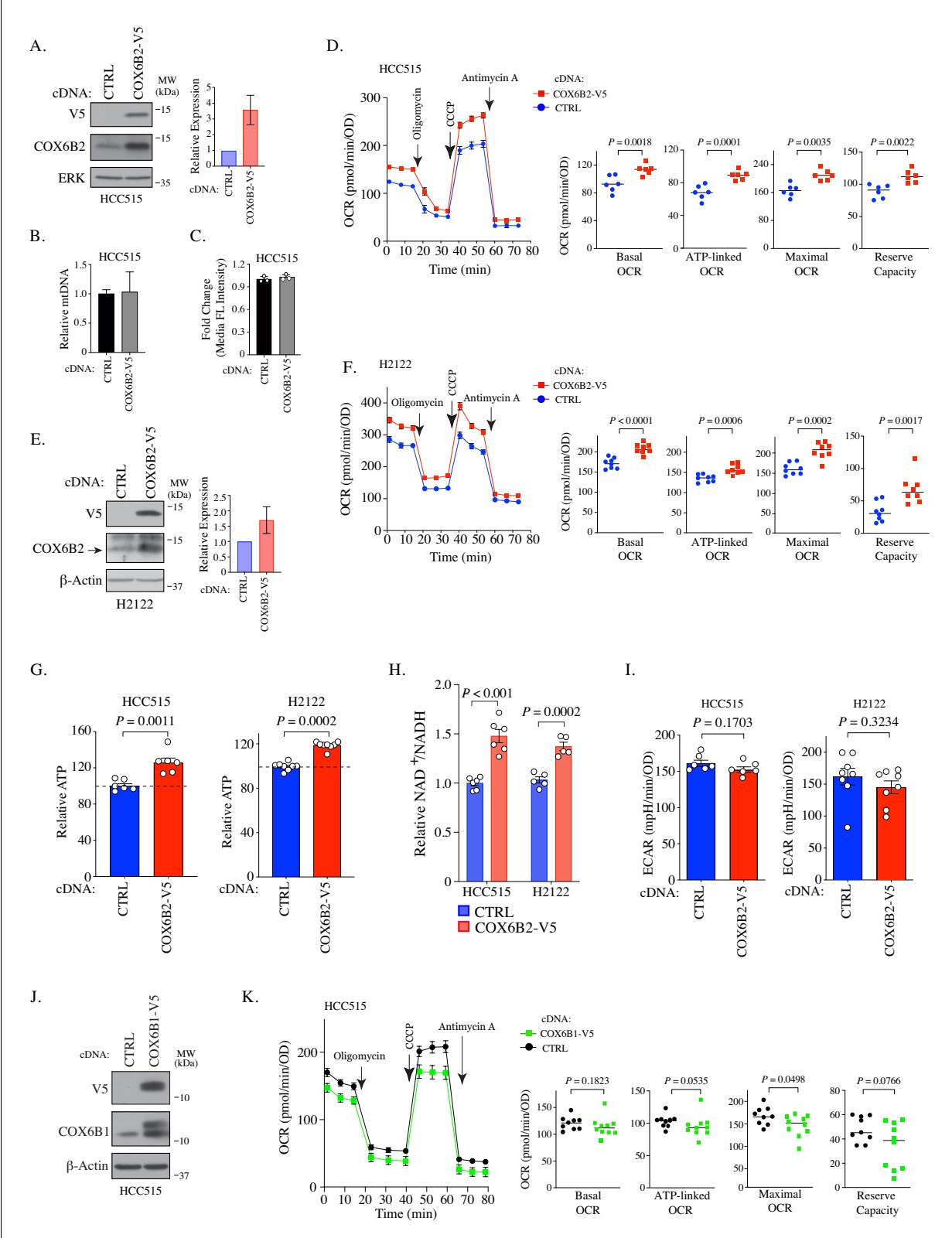

**Figure 2.** COX6B2 expression enhances mitochondrial respiratory activity. (**A**) Left: Whole cell lysates from HCC515 cells + / - COX6B2-V5 were immunoblotted with indicated antibodies. MW markers are indicated. Right: Quantitation of COX6B2 protein expression from blots on left. Bars represent mean ± SD (n = 4). (**B**) Mitochondrial DNA content. Bars represent mean + range (n = 2). (**C**) Quantification of total mitochondria using FACS analysis of Mitotracker green stained cells. Bars represent mean ± SD (n = 3). (**D**) Left: Oxygen consumption rate (OCR) as a function of time in

*Figure 2 continued on next page*

*Figure 2 continued*

indicated cell lines following exposure to electron transport chain complex inhibitors. Bars represent mean ± SEM (n = 6). Right: Mean and distribution of individual values for basal, ATP-linked, maximal and reserve OCRs. p-Values calculated by Student's *t*-test. (**E**) Left: Whole cell lysates from H2122 cells + / - COX6B2-V5 were immunoblotted with indicated antibodies. MW markers are indicated. Right: Quantitation of COX6B2 protein expression. Bars represent mean ± SD (n = 3). (**F**) As in (**D**) (n = 8). (**G**) ATP content of indicated cell lines. Bars represent mean + SEM (n ≥ 6). HCC515 p value calculated by Student's *t*-test and H2122 p value calculated by Mann-Whitney test. (**H**) NAD$^+$/NADH measurement in indicated cell lines. Bars represent mean ± SEM (n ≥ 5). p-Value calculated by Student's *t*-test. (**I**) Basal extracellular acidification rate (ECAR) are presented as mean ± SEM (n ≥ 6) from CTRL and COX6B2-V5 cells of HCC515 and H2122 cells. p-Value calculated by Student's *t*-test. (**J**) Immunoblots of indicated whole cell lysates with specified antibodies. MW markers are indicated. (**K**) As in (**D**) using indicated cell lines. Bars represent mean ± SEM (n ≥ 9). Basal and ATP-link OCR p value calculated by Student's *t*-test. Maximal OCR and reserve capacity p value calculated by Mann-Whitney test.

cell lines tested (*Figure 3B*). In contrast, a minimal, statistically insignificant increase in activity was observed following COX6B1-V5 overexpression (*Figure 3C*). The increase in complex IV activity did not correlate with any apparent increase in the abundance of complex IV as judged by COXIV accumulation (*Figure 3—figure supplement 1A*). This finding suggests that COX6B2 is a rate-limiting subunit of complex IV.

Recently, a number of reports have suggested that tissue-specific complex IV isoforms may promote supercomplex formation and selectively and specifically increase ATP production in tumor cells (*Maranzana et al., 2013*; *Ikeda et al., 2019*). Supercomplexes contain multiple electron transport sub-complexes containing complex III and IV and are hypothesized to promote local channeling of substrates, trapping of reactive intermediates and/or subunit stabilization (*Acín-Pérez et al., 2008*). To test this possibility for COX6B2, we used Blue-native PAGE (BN-PAGE) analysis to separate electron transport chain complexes (monomers (IV) and dimers (IV$_2$) and supercomplexes (III$_2$+IV and I +III$_2$+IV$_n$)) in control and COX6B1/B2-V5 expressing cells. Here, we observed that >90% of endogenous COX6B2 is found in in complex IV dimers or incorporated into supercomplexes. In contrast, >98% COX6B1 is found in monomeric complex IV (*Figure 3D–E*; *Figure 3—figure supplement 1B*). This trend is maintained when either isoform is overexpressed (>85% of COX6B2 is in complex IV in dimers/supercomplexes versus 40% for COX6B1) (*Figure 3F*).

We next tested the polymeric distribution of complex IV, using analysis of COXIV, a core protein subunit of complex IV, following COX6B1-V5 or COX6B2-V5 expression. Here, we found that upon expression of COX6B2-V5, but not COX6B1-V5, the distribution of complex IV shifted from monomers to increased formation of supercomplexes (*Figure 3G*). Supercomplexes are hypothesized to enhance electron transport activity without a compensatory increase in ROS (*Maranzana et al., 2013*). Indeed, COX6B2 overexpressing cells do not display a concomitant induction of superoxide (*Figure 3H*). These data indicate that COX6B2 preferentially integrates into dimeric complex IV and supercomplexes, and COX6B2 expression promotes incorporation of complex IV into supercomplexes. These alterations in distribution are proposed to enhance mitochondrial OXPHOS while limiting oxidative stress.

## COX6B2 enhances cell proliferation and confers a growth advantage in low oxygen

We next investigated whether the expression of COX6B2 is sufficient to enhance the growth of LUAD cells. We observed an increase in DNA synthesis as measured by EdU incorporation of 50.39 ± 6.35% in HCC515 and 38.7 ± 8.4% in H2122 (*Figure 4A*; *Figure 4—figure supplement 1A*). Notably, an increase is not observed for COX6B1-V5 overexpression consistent with no significant change in OCR (*Figure 4A*, middle panel; *Figure 2K*). In addition, we observed an enhanced population double rate of 37.16 ± 10.71% and 29.88 ± 6.74% for the HCC515 and H2122 cells, respectively (*Figure 4B* (-) Pyruvate column; compare red and blue bars; *Figure 4—figure supplement 1B*). Mitochondrial OXPHOS has been linked to proliferation via regulation of the NAD$^+$/NADH ratio, which is elevated in COX6B2-V5 overexpressing cells (*Figure 2H*; *Birsoy et al., 2015*; *Sullivan et al., 2015*). Consistent with this model, supplementation with pyruvate, an electron acceptor which drives NAD$^+$ production, largely diminishes the proliferative effects of COX6B2-V5 overexpression (*Figure 4B*). A similar proliferative advantage is also observed when COX6B2-V5 cells are grown in the absence of extracellular matrix, suggesting that COX6B2 promotes anchorage-independent growth (*Figure 4C*).

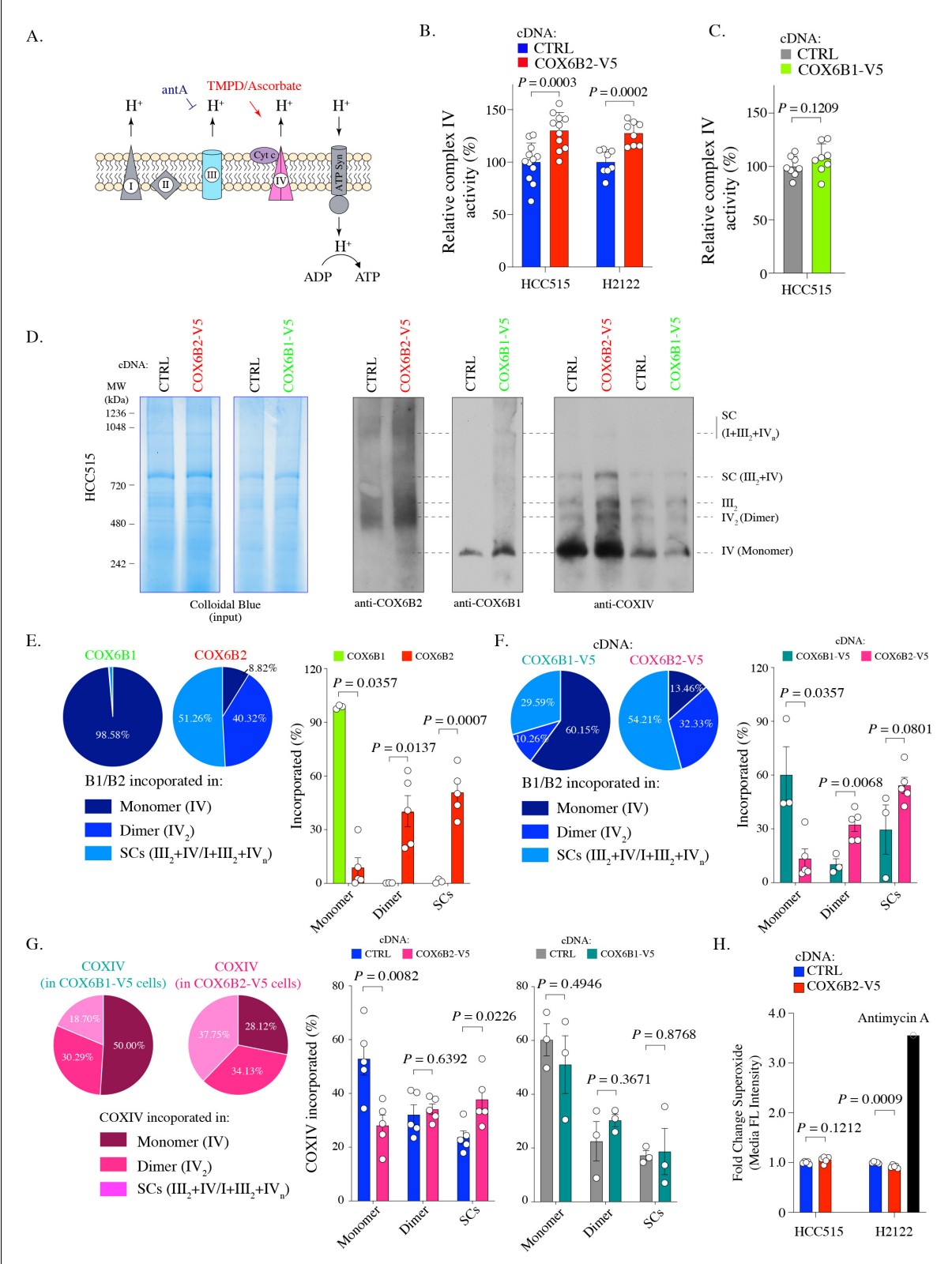

**Figure 3.** COX6B2 enhances complex IV activity of OXPHOS without increasing ROS production. (A) Schematic of complex IV activity measurements in isolated mitochondria. Details are described in Material and methods. antA, Antimycin A; Cyt c, Cytochrome c; ATP Syn, ATP synthase. (B) Complex IV activity in indicated cell lines using TMPD/ascorbate as a substrate. Bars represent mean + SEM (n ≥ 8). p-Value calculated by Student's *t*-test. (C) As in (B) in HCC515 cell lines expressing CTRL and COX6B1-V5 cDNA. Bars represent mean + SEM (n = 8). p-Value calculated by Student's *t*-test. (D)

*Figure 3 continued on next page*

*Figure 3 continued*

Indicated lysates from HCC515 cells were run on a BN-PAGE gel and stained with Colloidal Blue (left two panels) or immunoblotted with anti-COX6B2, anti-COX6B1 and anti-COXIV (right panels). Representative image of n ≥ 3. MW markers are indicated to identify different complexes (*Mourier et al., 2014*). (E) Left: Distribution of COX6B1 and COX6B2 incorporated in complex IV monomers, dimers or supercomplexes as detected by BN-PAGE in (D). Right: Bars represent mean ± SEM (n ≥ 3) based on quantification of bands in (D). In monomeric complex IV, p value calculated by Mann-Whitney test whereas others calculated by Student's *t*-test. (F) Left: Distribution of COX6B1/COX6B2 incorporated in monomeric complex IV, dimeric complex IV and supercomplexes based on BN-PAGE in COX6B2-V5 or COX6B1-V5 overexpressing cell lines (D). Right: Bars represent mean ± SEM (n ≥ 3) based on quantification of bands in (D). In monomeric complex IV, p value calculated by Mann-Whitney test whereas others calculated by Student's *t*-test. (G) Left: Distribution of COXIV incorporated in monomeric complex IV, dimeric complex IV and supercomplexes by blots from (D). Right: Bars represent mean ± SEM (n ≥ 3) of COXIV based on quantification of band in (D). p-Value calculated by Student's *t*-test. (H) Relative superoxide in indicated cell lines. Bars represent mean + SEM (n = 5). p-Value calculated by Student's *t*-test.

The online version of this article includes the following figure supplement(s) for figure 3:

**Figure supplement 1.** Measurement of complexes and supercomplexes.

Isoforms of complex IV subunits can be selectively expressed to permit adaptation to tissue-specific needs or environmental conditions (*Sinkler et al., 2017*). In particular, hypoxia is reported to regulate complex IV subunit expression and permit continued respiration in low oxygen (*Ikeda et al., 2019*; *Hayashi et al., 2015*; *Fukuda et al., 2007*). Therefore, we cultured control and COX6B2-V5 overexpressing cells in low (2.5%) $O_2$ for 4 days, monitoring EdU incorporation every 24 hr. Under these conditions, control cells reduce EdU incorporation by ~60%, while COX6B2-V5 expressing cells decrease incorporation by 15% (*Figure 4D*; *Figure 4—figure supplement 1C*). This data indicates that COX6B2 expression can enhance proliferation in both normoxia and low oxygen conditions. Given this phenotype, we examined whether COX6B2 is regulated in response to low oxygen. ChIP data from human breast cancer cells indicates that HIF-1 is bound near the *COX6B2* promoter (*Zhang et al., 2015*). In addition, we also identified 11 putative HREs (A/G-CGTG) which are enriched at the HIF-1 peaks (*Figure 4—figure supplement 1D*). Indeed, we observed a greater than twofold increase in *COX6B2* but not *COX6B1* mRNA following 12 hr of 2.5% oxygen (*Figure 4E*). This accumulation of mRNA was associated with stabilization of COX6B2 protein (*Figure 4F*). Under both normoxic or hypoxic conditions, we did not observe any significant alteration in COX6B2 or COX6B1 accumulation when the alternative isoform was overexpressed, indicating that the changes in expression are not an isoform switch (*Figure 1—figure supplement 1B*; *Figure 4—figure supplement 1E*). Collectively, these data suggest that the enhanced electron transport activity in COX6B2 expressing cells confers a selective advantage by promoting key tumorigenic behaviors including unrestrained proliferation, anchorage independent growth and enhanced survival in hypoxia.

## Depletion of COX6B2 impairs mitochondrial electron transport chain function in LUAD cells

Given the pro-proliferative phenotype following COX6B2 expression, we next examined the consequences of its depletion on OXPHOS and tumor cell viability. Reduction of COX6B2 protein using shRNA pools (four independent sequences) or siRNA pools (two independent sequences), led to a decrease in basal, ATP-linked, maximal and reserve OCR in HCC515 and H2122 cell lines (*Figure 5A–B*; *Figure 5—figure supplement 1A*). We were unable to establish a stable clone of COX6B2 deleted cells by CRISPR/Cas9. However, COX6B2 directed CRISPR/Cas9 did generate a population with reduced COX6B2 protein expression (*Figure 5—figure supplement 1B*). In agreement with the si and shCOX6B2 data, sgCOX6B2 cells also exhibit attenuated OCR (*Figure 5—figure supplement 1C*). The reduction in OCR following COX6B2 depletion is accompanied by a reduction in the NAD$^+$/NADH ratio (*Figure 5C*). In addition, total cellular ATP is decreased following siRNA-mediated COX6B2 depletion with two independent siRNA sequences across a panel of COX6B2-expressing LUAD cell lines (*Figure 5D*). Consistent with a loss of electron transport chain function, depletion of COX6B2 resulted in decreased mitochondrial membrane potential (*Figure 5E*) and increased intracellular hydrogen peroxide (*Figure 5F*). Collectively, these findings indicate that COX6B2 is essential for respiration and mitochondrial integrity in tumor cells.

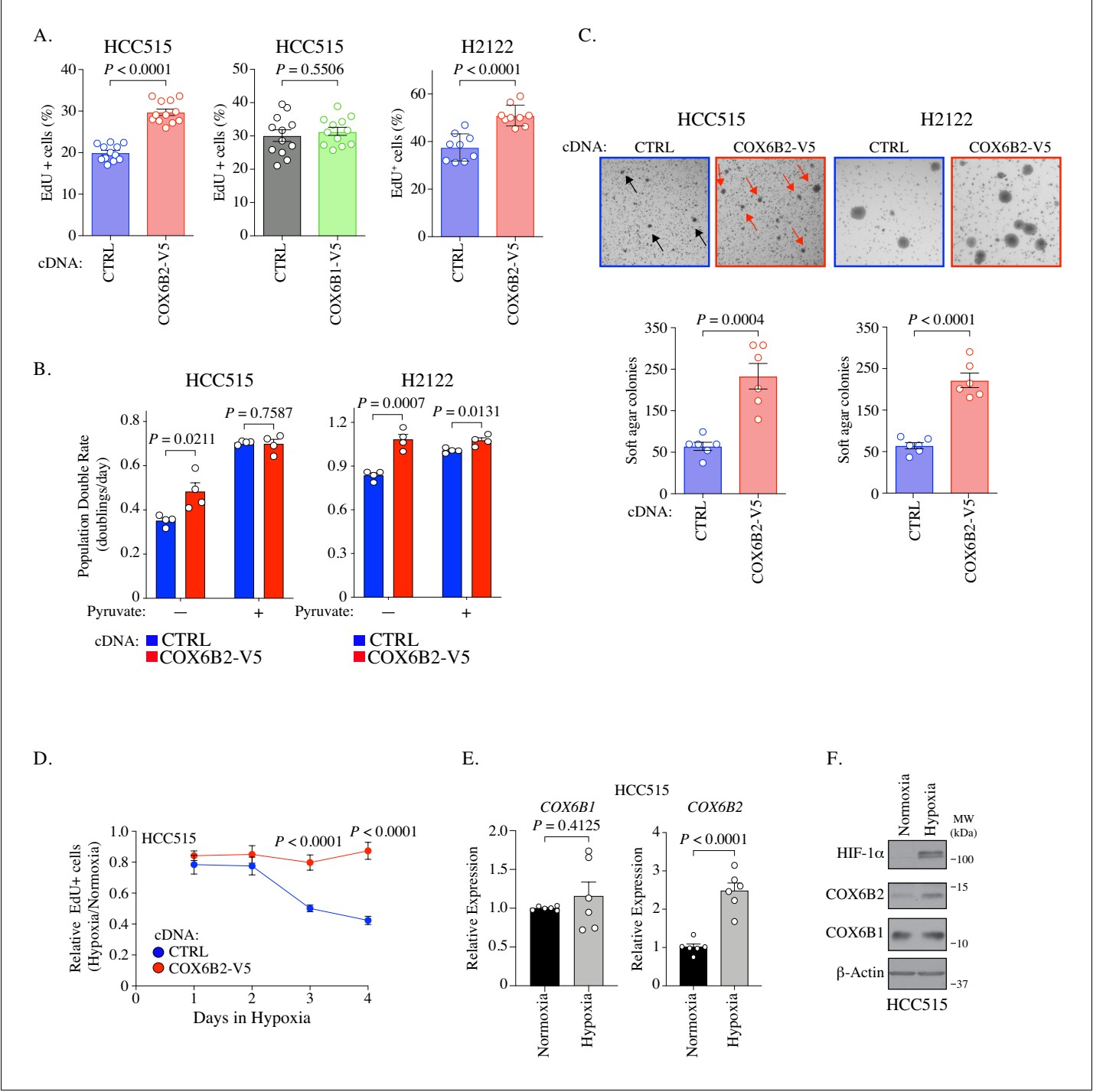

**Figure 4.** COX6B2 promotes cell division. (A) Percentage of cells positive for EdU incorporation in indicated samples. Values represent the mean ± SEM (n ≥ 9). p-Value calculated by Student's *t*-test. (B) Population doubling rate in indicated samples incubated ±1 mM pyruvate for 5 days. Bars represent mean + SEM (n = 4). p-Value calculated by Student's *t*-test. (C) Top: Representative images of soft agar assays for indicated cell lines. Arrows indicate formed colonies. Bottom: Graphs represent mean colony numbers in indicated cell lines. Bars represent mean ± SEM (n = 6). p-Value calculated by Student's *t*-test. (D) EdU-positive HCC515 cells at indicated times in hypoxia. Values are relative to cells incubated in normoxia. Bars represent mean ± SEM (n = 8). p-Value calculated by Student's *t*-test. (E) Average mRNA expression of *COX6B1* and *COX6B2* measured following 12 hr of hypoxia culture in HCC515 cells. Bars represent mean ± SEM (n = 6). P calculated by Student's *t*-test. (F) Whole cell lysates from indicated cell lines were immunoblotted with indicated antibodies. Hypoxia exposure was for 12 hr. Representative immunoblots from n = 3. MW markers are indicated. The online version of this article includes the following figure supplement(s) for figure 4:

**Figure supplement 1.** COX6B2 promotes cell division.

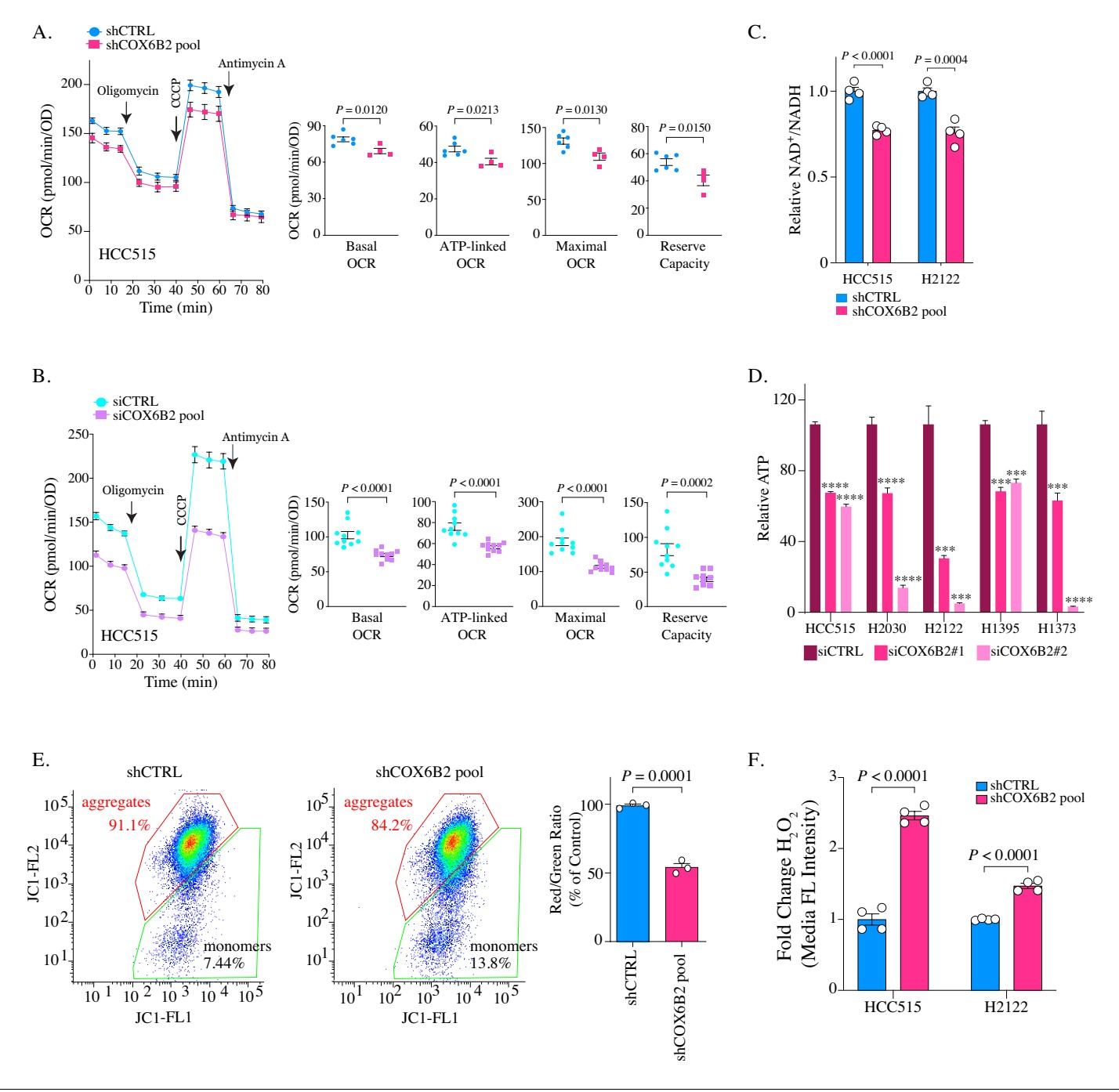

**Figure 5.** Depletion of COX6B2 impairs mitochondrial function. (**A–B**) Left: Oxygen consumption rate (OCR) as a function of time in COX6B2 depleted HCC515 cells using an shRNA pool (**A**) or an siRNA pool (**B**) following exposure to electron transport chain complex inhibitors. Bars represent mean ± SEM (n ≥ 4). Right: Bars represent mean ± SEM 20 min following the addition of each drug (on left). p-Values calculated by Student's t-test. (**C**) Relative NAD$^+$/NADH measurement in indicated cell lines. Bars represent mean + SEM (n = 4). p-Value calculated by Student's t-test. (**D**) Relative ATP content in indicated cell lines transfected with indicated individual siRNAs for 96 hr. Bars represent mean + SEM (n = 6), ****p<0.0001; ***p<0.005 calculated by Student's t-test. (**E**) Left panels: Flow cytometry measurement following JC-1 labelling of indicated HCC515 cells. Right graph: Bars represent mean of the aggregate to monomer ratio (red/green) in indicated cell lines + SEM (n = 3). p-Value calculated by Student's t-test. (**F**) Indicated cell lines were labelled with CM-H2DCFDA to measure H$_2$O$_2$ as described in Material and methods. Bars represent mean fold change in median fluorescent intensities ± SEM (n = 4). p-Value calculated by Student's t-test.

The online version of this article includes the following figure supplement(s) for figure 5:

**Figure supplement 1.** Depletion of COX6B2 impairs OXPHOS.

## Depletion of COX6B2 decreases cell survival and induces cell apoptosis and senescence

Based on the observed decrease in OCR, we asked whether tumor cell viability was diminished following COX6B2 depletion in LUAD cells. Depletion of COX6B2 for nine days, results in a decrease in EdU incorporation (*Figure 6A*). Furthermore, these cells are severely compromised in their ability to grow in the absence of an extracellular matrix (*Figure 6B*). Both of these phenotypes were recapitulated in sgCOX6B2 cells (*Figure 6—figure supplement 1A–B*). We next examined the underlying mechanism(s) leading to the observed loss of viability. In our original screen, we found that siRNA-mediated depletion of COX6B2 enhances cleavage of caspase 3/7, a phenotype that is rescued in COX6B2-V5 overexpressing cells (*Figure 6C*, *Figure 6—figure supplement 1C*; *Maxfield et al., 2015*). In cells where COX6B2 is depleted using a pool of shRNAs, with sequences independent of the those in the siRNA pool, we observed cleavage of caspase-3 and PARP (*Figure 6C*). These cell death markers are also induced in sgCOX6B2 cells (*Figure 6—figure supplement 1D*). The induction of cell death proteins is accompanied by an increase in Annexin-V staining, demonstrating activation of caspase-dependent programmed cell death (*Figure 6D*). Visual inspection of COX6B2-depleted cultures also revealed a population of cells that are flattened, accumulate vacuoles, and in some cases, are multi-nucleated, feature reminiscent of senescence (*Figure 6E*). Staining of these cells revealed an increase in β-galactosidase, suggesting induction of a cellular senescence program (*Figure 6F*). To determine the underlying trigger for the reduction in survival, we attempted to rescue the death through multiple approaches. Supplementing depleted cells with pyruvate and uridine as metabolic and nucleotide fuel was insufficient to rescue the viability defects (*Figure 6—figure supplement 1E–F*). However, exposing cells to the ROS scavenger N-acetyl-l-cysteine (NAC) reduced the extent of the cleaved caspase-3 and PARP and β-galactosidase staining (*Figure 6G–H*). Thus, COX6B2 is essential for survival of LUAD tumor cells as its suppression reduces ATP, compromises mitochondrial integrity thereby generating excess ROS that leads to apoptosis or senescence.

## COX6B2 is necessary and sufficient for tumor growth in vivo

We next sought to determine whether COX6B2 was necessary and/or sufficient for tumor growth in vivo. We implanted control and COX6B2-V5 HCC515 cells into the flank of FOXn1$^{nu}$ (nude) mice. Importantly, tumors engrafted subcutaneously encounter a highly hypoxic environment (~0.08–0.8% $O_2$) (*Helmlinger et al., 1997*). In agreement with the cell line data, we observed enhanced growth of COX6B2-V5 expressing cells in vivo (*Figure 7A*). After 38 days, COX6B2-V5 tumors exhibit a 4X greater mass than control (*Figure 7B*). In the converse experiment, we xenografted control and shCOX6B2 cells in the flank of immunocompromised mice. Complementing the gain-of-function phenotype observed, we found that depletion of COX6B2 severely reduces tumor growth and final tumor mass (*Figure 7C–D*). Taken together, our findings indicate that the anomalous expression of COX6B2 is both necessary and sufficient for growth of LUAD tumors.

## Discussion

COX6B2 function has remained obscure in both tumor cells and sperm since its discovery (*Hüttemann et al., 2003*). Our data indicate that COX6B2 enhances complex IV activity leading to an increase in mitochondrial oxidative phosphorylation. The function of COX6B2 appears to differ from COX6B1, which did not enhance OXPHOS when overexpressed, a finding consistent with previous biochemical analyses that indicate removal of COX6B1 enhances complex IV activity (*Weishaupt and Kadenbach, 1992*). Genetic studies demonstrate that human mutations in COX6B1 reduce assembly of complex IV and other ETC complexes. Based on this cumulative data, we hypothesize that COX6B1 (and perhaps B2) are essential for assembly and regulation of complex IV. In normal tissues, COX6B1 may regulate complex IV to prevent the generation of excessive and unused energy and also damaging ROS. In sperm, where ATP demand is high, COX6B2 expression could promote complex IV dimerization and incorporation into supercomplexes to promote efficient OXPHOS and limit ROS production that would otherwise damage DNA (*Agarwal et al., 2014*). This function may be particularly important during early tumorigenesis, where a highly hypoxic environment can lead to proliferative arrest and excessive ROS generation. We propose that tumor cells

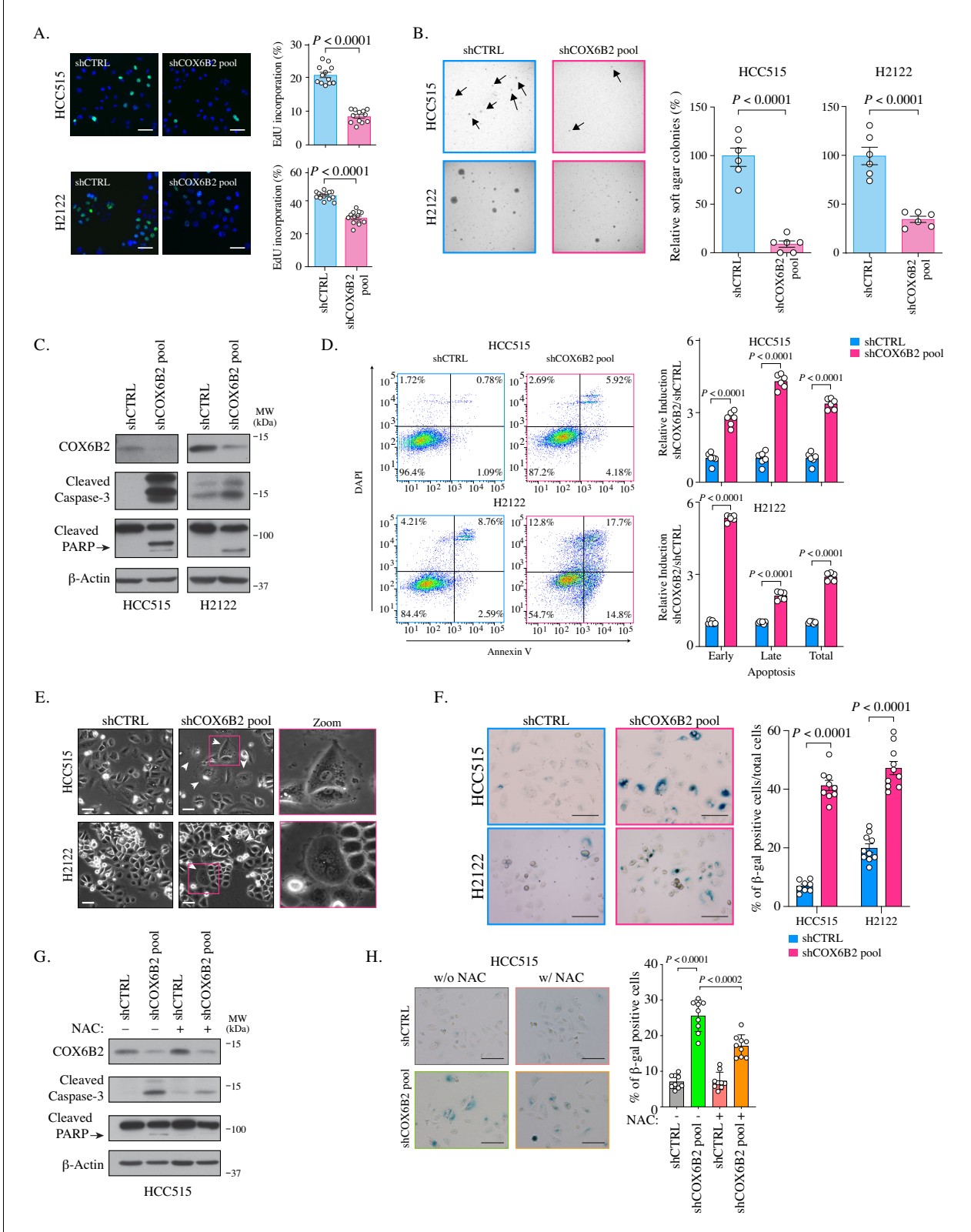

**Figure 6.** COX6B2 is essential for tumor cell viability and survival. (A) Left: Representative images of EdU staining in indicated cell lines. (EdU, green; Hoechst, blue). Scale bar, 100 μm. Right: Quantitation of EdU-positive cells. Bars represent the mean ± SEM (n ≥ 12). p-Value calculated by Student's *t*-test. (B) Left: Representative images of colonies in soft agar. Arrows indicate formed colonies. Right: Bars represent relative mean colony numbers ± SEM (n = 6). p-Value calculated by Student's *t*-test. (C) Representative immunoblots of whole cell lysates of indicated cells blotted with

*Figure 6 continued on next page*

*Figure 6 continued*

indicated antibodies. MW markers are indicated. (D) Left: Representative scatter plots of Annexin V staining in indicated cell lines. Bars represent mean fold change + SEM (n = 6). p Calculated by Student's *t*-test. (E) Representative phase contrast images of indicated cell lines. Arrowheads indicate the flattened senescence morphology with vacuole accumulation and multinucleation. Scale bar, 100 μm. (F) Representative images of senescence-associated- β-galactosidase staining in indicated cells. Scale bar, 100 μm. Bars represent mean ± SEM (n ≥ 8). p Calculated by Student's *t*-test. (G) Whole cell lysates from indicated cell lines were cultured with or without NAC (5 mM) for 7 days were immunoblotted with indicated antibodies. MW markers are indicated. (H) Representative images of senescence-associated- β-galactosidase staining in indicated cell lines cultured with or without NAC (5 mM) for all 11 days. Scale bar, 100 μm. Bars represent mean ± SEM (n = 9). p Calculated by Student's *t*-test.

The online version of this article includes the following figure supplement(s) for figure 6:

**Figure supplement 1.** Phenotypes following COX6B2 depletion.

---

that express and engage the COX6B2-based, sperm-specific mechanism can overcome this innate barrier to transformation that would otherwise restrict cell survival.

One of the most surprising aspects of our study was the loss of tumor cell viability following the depletion of COX6B2. As COX6B2 enhances OXPHOS, its loss would be expected to reduce ATP levels and the $NAD^+$/NADH ratio and slow proliferation, but not necessarily kill tumor cells. Tumor

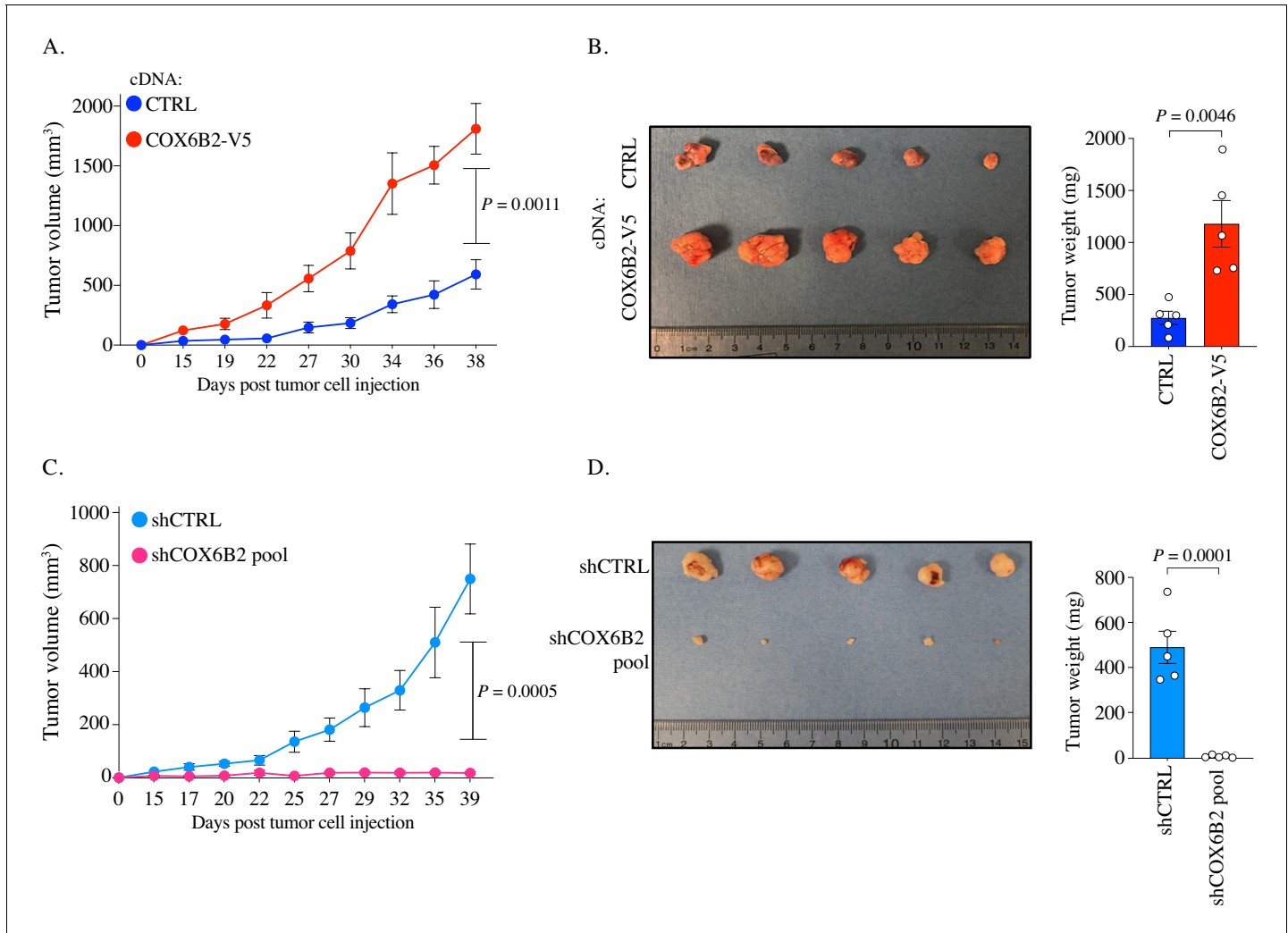

**Figure 7.** COX6B2 supports tumor growth in vivo. (A and C) Xenograft experiments in indicated cell lines. Tumor volumes were measured using calipers on indicated days. Each data point represents mean ± SEM (n = 5). p-Value calculated by Student's *t*-test. (B and D) Left: Images of individual tumors (ruler indicates cm). Right: Mass of excised tumors from indicated cell lines. Bars represent mean ± SEM (n = 5). p-Value calculated by Student's *t*-test.

cell glycolytic pathways should theoretically compensate for reduced energy generation. Moreover, the presence of COX6B1 should be able to support complex IV function. Instead, we observe ROS production and an exit from the cell cycle either to death or senescence. This finding implies that these cells are highly dependent upon OXPHOS for survival. In addition, the loss of COX6B2 may promote apoptosis due to a collapse of mitochondrial membrane potential and cytochrome c egress (*Smaili et al., 2001*; *Tai et al., 2017*).

The majority of tumor metabolism studies have focused on the contribution of the Warburg effect and have led to the notion that OXPHOS is dispensable for many transformed cells. However, a growing body of evidence indicates that OXPHOS is essential for tumor cell division and survival. First, OXPHOS contributes to >80% of total ATP production in tumor cells, which is essential for satisfying the demands of proliferation and tumor progression (*Rao et al., 2019*; *Lissanu Deribe et al., 2018*; *Zu and Guppy, 2004*). Second, OXPHOS provides electron acceptors to generate metabolic intermediates (e.g. aspartate) that are required for continued cell division (*Sullivan et al., 2015*). Third, tumor cells exhibit enhanced sensitivity to inhibitors of OXPHOS (*Nayak et al., 2018*; *Scotland et al., 2013*). Fourth, isoform switching of complex IV subunits is observed in tumors and is associated with aggressive malignant behaviors (*Oliva et al., 2015*; *Zhang et al., 2016*; *Minton et al., 2016*). Finally, OXPHOS is upregulated and essential for survival during ER-stress, hypoxia and glucose limiting conditions, and exposure to targeted therapies (*Balsa et al., 2019*; *Ikeda et al., 2019*; *Fukuda et al., 2007*; *Guièze et al., 2019*). Based on these and a vast array of additional data, it is accepted that inhibition of OXPHOS is a tractable therapeutic entry point for treating cancer (*Weinberg and Chandel, 2015*). Indeed, extensive studies with metformin have demonstrated anti-proliferative activities and reduction of tumor incidence through the inhibition of Complex I (*Weinberg and Chandel, 2015*). However, this and other ETC inhibitors cannot discriminate between tumor or normal cells, limiting their therapeutic window. We propose that enhanced accumulation of COX6B2 is an acquired trait that is essential for OXPHOS in LUAD and perhaps additional cancers. At minimum, elevated COX6B2 protein expression is a functional enrollment biomarker for OXPHOS-inhibitor-based treatment. Moreover, COX6B2 structural elements that are distinct from COX6B1 could represent targets for small molecule inhibitors. Finally, due to expression in an immune privileged site, CTAs have long been considered ideal targets for immunotherapy. However, no objective criterion exists to prioritize them for immunotherapy. We propose that COX6B2 represents an ideal candidate for immunotherapy because it has a bona fide tumorigenic function. Specifically, sequence differences between COX6B2 and COX6B1 could permit antigen recognition by patient immune cells and thus act as targets of adoptive T-cell transfer.

## Materials and methods

**Key resources table**

| Reagent type (species) or resource | Designation | Source or reference | Identifiers | Additional information |
|---|---|---|---|---|
| Strain, strain background (*Mus musculus*, female) | Athymic nude mice | Envigo | Hsd:Athymic Nude-Foxn1[nu] | |
| Cell line (*Homo sapiens*) | HEK293T | ATCC | CRL-3216; RRID:CVCL_0063 | |
| Cell line (*Homo sapiens*) | H1395 | John Minna | RRID:CVCL_1467 | UT Southwestern |
| Cell line (*Homo sapiens*) | H1373 | John Minna | RRID:CVCL_1465 | UT Southwestern |
| Cell line (*Homo sapiens*) | H2030 | John Minna | RRID:CVCL_1517 | UT Southwestern |
| Cell line (*Homo sapiens*) | H2122 | John Minna | RRID:CVCL_1531 | UT Southwestern |
| Cell line (*Homo sapiens*) | H2126 | John Minna | RRID:CVCL_1532 | UT Southwestern |

*Continued on next page*

*Continued*

| Reagent type (species) or resource | Designation | Source or reference | Identifiers | Additional information |
|---|---|---|---|---|
| Cell line (*Homo sapiens*) | H1944 | John Minna | RRID:CVCL_1508 | UT Southwestern |
| Cell line (*Homo sapiens*) | HCC515 | John Minna | RRID:CVCL_5136 | UT Southwestern |
| Cell line (*Homo sapiens*) | HCC1195 | John Minna | RRID:CVCL_5127 | UT Southwestern |
| Antibody | Anti-COX6B2 (Rabbit polyclonal) | MilliporeSigma | Cat# SAB1401983, RRID:AB_10609550 | IB (1:1000) IF (1:50) IHC (1:150) |
| Antibody | Anti-COX6B1 (Mouse monoclonal) | Santa Cruz Biotechnology | Cat# sc-393233, RRID:AB_2814984 | IB (1:1000) |
| Antibody | Anti-COXIV (Rabbit monoclonal) | Cell Signaling Technology | Cat# 4850, RRID:AB_2085424 | IB (1:5000) IF (1:200) |
| Antibody | Anti-β-Actin (Mouse monoclonal) | Santa Cruz Biotechnology | Cat# sc-47778 | IB (1:10,000) |
| Antibody | Anti-V5 (Mouse monoclonal) | Thermo Fisher Scientific | R960-25; RRID:AB_2556564 | IB (1:5000) |
| Antibody | Anti-HIF-1α (Mouse monoclonal) | BD Biosciences | Cat# 610959, RRID:AB_398272 | IB (1:1000) |
| Antibody | Anti-Cleaved Caspase-3 (Rabbit polyclonal) | Cell Signaling Technology | Cat# 9661, RRID:AB_2341188 | IB (1:500) |
| Antibody | Anti-PARP (Rabbit monoclonal) | Cell Signaling Technology | Cat# 9532, RRID:AB_659884 | IB (1:1000) |
| Antibody | Anti-ERK (Mouse monoclonal) | Santa Cruz Biotechnology | Cat# sc-135900, RRID:AB_2141283 | IB (1:3000) |
| Antibody | Anti-NDUFA9 (Mouse monoclonal) | Thermo Fisher Scientific | Cat# 459100, RRID:AB_2532223 | IB (1:1000) |
| Antibody | Anti-UQCRC2 (Mouse monoclonal) | Abcam | Cat# ab14745, RRID:AB_2213640 | IB (1:1000) |
| Antibody | Anti-Tom20 (Mouse monoclonal) | Santa Cruz Biotechnology | Cat# sc-17764, RRID:AB_628381 | IF (1:200) |
| Commercial assay or kit | MitoTracker Green | Thermo Fisher Scientific | M7514 | |
| Commercial assay or kit | MitoSOX Red | Thermo Fisher Scientific | M36008 | |
| Commercial assay or kit | MitoProbe JC-1 | Thermo Fisher Scientific | M34152 | |
| Commercial assay or kit | Cell-Titer Glo | Promega | PR-G7572 | |
| Commercial assay or kit | NAD$^+$/NADH-Glo Assay | Promega | G9071 | |
| Commercial assay or kit | Click-iT EdU Alexa Fluor 488 Imaging Kit | Thermo Fisher Scientific | C10037 | |
| Commercial assay or kit | Senescence Beta-Galactosidase Staining Kit | Cell Signaling Technology | 9860 | |
| Commercial assay or kit | Colloidal Blue Staining Kit | Thermo Fisher Scientific | | |
| Chemical compound, drug | oligomycin | MilliporeSigma | O4876 | |
| Chemical compound, drug | CCCP | MilliporeSigma | C2759 | |
| Chemical compound, drug | Antimycin A | MilliporeSigma | A8674 | |

*Continued on next page*

*Continued*

| Reagent type (species) or resource | Designation | Source or reference | Identifiers | Additional information |
|---|---|---|---|---|
| Chemical compound, drug | Seahorse XF Plasma Membrane Permeabilize | Agilent | 102504–100 | |
| Chemical compound, drug | NAC | MilliporeSigma | N7250 | |
| Chemical compound, drug | Saponin | MilliporeSigma | 47036 | |
| Software, algorithm | FlowJo | FlowJo | RRID:SCR_008520 | ver 10 |
| Software, algorithm | ImageJ (Fiji) | *Schindelin et al., 2012* | RRID:SCR_002285 | ver 2.0.0 |
| Software, algorithm | Integrative Genomics Viewer | *Robinson et al., 2011* | RRID:SCR_011793 | ver 2.3.93 |

## Cell lines

All NSCLC cell lines were obtained from John Minna (UT Southwestern) between 2014 and 2018. NSCLC cells were cultured in RPMI media supplemented with 5% FBS at 37°C, 5% $CO_2$ and 90% humidity. NSCLC cells were not passaged more than 10 times after thawing. Cells cultured under hypoxia followed the methods described previously (*Wright and Shay, 2006*), and each chamber attained oxygen concentration of ~2.5%. HEK293T cells were obtained from ATCC in 2014 and cultured in DMEM media supplemented with 10% FBS at 37°C, 5% $CO_2$ and 90% humidity. HEK293T cells were not passaged more than three times after thawing. All cells were authenticated between 2014 and 2020 using short tandem repeat profiling and periodically evaluated for mycoplasma contamination by DAPI stain for extra-nuclear DNA within one year of use.

## Reagents

Chemicals and reagents were purchased from the following manufacturers: RPMI-1640 medium (R8758), DMEM (D6429), fetal bovine serum (FBS; F0926, batch# 17J121), HBSS (H8264), EGTA (E3889), fatty acid free BSA (A7030), oligomycin (O4876), CCCP (C2759), antimycin A (A8674), mannitol (M4125), $KH_2PO_4$ (P0662), $MgCl_2$ (M8266), ADP-$K^+$ salt (A5285), ascorbate (A5960), $N,N,N',N'$-Tetramethyl-$p$-phenylenediamine dihydrochloride (TMPD; 87890), sodium pyruvate solution (S8636), L-glutamine (G7513), dodecyltrimethylammonium bromide (DTBA; D8638), DAPI (D9542), saponin (47046), (uridine (U3003), and $N$-Acetyl-L-cysteine (NAC, N7250) were purchased from MilliporeSigma. HEPES (BP410), sucrose (50-712-768), EDTA (S311), Tris Base (BP152), formaldehyde (BP521), and Hoechst 33342 (PI62249) were purchased from Fisher Scientific. Seahorse XF Plasma Membrane Permeabilizer (XF PMP; 102504–100) are from Agilent.

## Normal lung lysate and human LUAD tissues

Human normal lung whole tissue lysate was purchased from Novus Biologicals (NB820-59237). All human LUAD tissues were obtained from the UTSW Tissue Resource in compliance with guidelines for informed consent approved by the UTSW Internal Review Board committee. Samples were homogenized in RIPA buffer (50 mM Tris-HCl pH7.4, 150 mM NaCl, 1% Triton X-100, 0.5% Sodium Deoxycholate, 0.1% SDS, 1 mM EDTA, 10 mM NaF, 1 µg ml$^{-1}$ pepstatin, 2 µg ml$^{-1}$ leupeptin, 2 µg ml$^{-1}$ aprotinin and 50 µg ml$^{-1}$ bestatin). The protein was recovered by centrifuge and quantitated by Pierce BCA Protein Assay Kit (23227, Thermo Fisher Scientific).

## Kaplan-Meier analysis

Overall survival and time to first progression graphs were generated using kmplotter (http://www.kmplot.com/lung) (*Győrffy et al., 2013*). In NSCLC, probability of overall survival (OS) was based on 1144 patients and time to first progression (FP) in 596 patients. In lung adenocarcinoma (LUAD), OS was based on 672 patients and FP on 443 patients. In lung squamous carcinoma (LUSC), OS was based on 271 patients and FP on 141 patients. Hazard ratios and p-values were calculated by Cox Regression Analysis.

## TCGA data analysis

We used the UCSC Xena program (https://xenabrowser.net) to download RNA-Seq data from the cohort of TCGA Lung Cancer (LUNG). Values were analyzed using Prism Graphpad software (v 8.4.3).

## Immunoblotting

SDS-PAGE and immunoblotting were performed as previously described (*Maxfield et al., 2015*). Antibodies used for immunoblotting were as follows: COX6B2 (1:1000, SAB1401983, Millipore-Sigma), COX6B1 (1:1000, sc-393233, Santa Cruz Biotechnology), COXIV (1:5000, 4850, Cell Signaling Technology), β-Actin (1:10,000, sc-47778, Santa Cruz Biotechnology), V5 (1:5000, R960-25, Thermo Fisher Scientific), HIF-1$\alpha$ (1:1000, 610959, BD Biosciences), Cleaved Caspase-3 (1:500, 9661, Cell Signaling Technology), PARP (1:1000, 9532, Cell Signaling Technology), ERK (1:3000, sc-135900, Santa Cruz Biotechnology), NDUFA9 (1:1000, 459100, Thermo Fisher Scientific), UQCRC2 (1:1000, ab14745, Abcam). Proteins were quantitated by ImageJ software and normalized to CTRL groups. Incorporated COX6B1, COX6B2 or COXIV in individual complex (monomers (IV) and dimers (IV$_2$) and supercomplexes (III$_2$+IV and I+III$_2$+IV$_n$)) was normalized to total complexes (sum of IV, IV$_2$, III$_2$+IV and I+III$_2$+IV$_n$).

## Immunofluorescence

Cells plated on glass coverslips were washed with PBS, fixed with 4% formaldehyde for 15 min at room temperature and blocked for 3 hr at room temperature in blocking buffer (5% BSA in PBS with 0.1% Saponin). Cells were incubated with primary antibodies diluted with blocking buffer overnight at 4°C followed by washes and incubation with Alexa Fluor-conjugated secondary antibodies (Thermo Fisher Scientific) for 1 hr at room temperature. ProLong Gold Antifade reagent with DAPI (Thermo Fisher Scientific) was used to mount slips on glass slides and images were acquired by Zeiss LSM510 confocal microscope or Leica DM5500 B upright microscope. Antibodies used: COX6B2 (1:50, SAB1401983, MilliporeSigma), Tom20 (1:200, sc-17764, Santa Cruz Biotechnology), COXIV (1:200, 4850, Cell Signaling Technology). Staining was quantitated using ImageJ software.

## Human lung tissue immunohistochemistry (IHC)

Non-malignant testis, lung and LUAD tissues were obtained with informed consent from the UTSW Tissue Management Shared Resource (TMSR) in compliance with the UTSW Internal Review Board committee. COX6B2 IHC was optimized and performed by the TMSR according to internal protocols using the Dako Autostainer Link 48 system. Briefly, the slides were baked for 20 min at 60°C, then deparaffinized and hydrated before the antigen retrieval step. Heat-induced antigen retrieval was performed at low pH for 20 min in a Dako PT Link. The tissue was incubated with a peroxidase block and then an antibody incubated (COX6B2; 1:150, SAB1401983, MilliporeSigma) for 35 min. The staining was visualized using the EnVision FLEX visualization system.

Staining scores of COX6B2 ranging from 0 to 3 were set based on the positive scores quantitated by using IHC Profiler of ImageJ (*Varghese et al., 2014*): score 0 (<30), score 1 (30–45), score 2 (45-60), and score 3 (>60). *Figure 1G* demonstrates a 0 for negative staining, whereas score 3 shows the highest staining intensity.

Expression plasmids cDNAs encoding human *COX6B2* and *COX6B1* were obtained in pDONR223 (DNASU) and cloned into pLX302 and pLX304 respectively using the Gateway Cloning system (Thermo Fisher Scientific). psPAX2 and pMD2.G lentiviral constructs were purchased from Addgene. For shRNA experiments, pLKO.1 vectors from TRC expressing COX6B2-targeted shRNAs (TRCN0000046119-122; sequences: 5'-CCGGGAGCAGATCAAGAACGGGATTCTCGAGAATCCCG TTCTTGATCTGCTCTTTTTG-3', 5'-CCGGGATCCGTAACTGCTACCAGAACTCGAGTTCTGGTAG-CAGTTACGGATCTTTTTG-3', 5'-CCGGCTACCAGA ACTTCCTGGACTACTCGAGTAGTCCAGGAAG TTCTGGTAGTTTTTG-3', 5'-CCGGCC AGCCAGAACCAGATCCGTACTCGAGTACGGATCTGGTTC TGGCTGGTTTTTG-3') were used as shRNA pool. Nontargeting shRNA in pLKO.1 (shSCR) was used as a control (Addgene). For sgRNA experiments, pSpCas9(BB)−2A-GFP (PX458) was purchased from Addgene.

## Lentiviral transduction

Lentivirus was produced through co-transfection of HEK293T cells with 5 µg of viral expression vector, 3 µg of psPAX2 packing vector, and 2 µg of pMD2.G envelope vector. Virus-conditioned medium was harvested, passed through 0.45-µm-pore-size filters, and then used to infect target cells in the presence of 10 µg/ml Sequa-brene (S2667, MilliporeSigma).

## Mitochondrial content and mass analysis

Mitochondrial DNA quantification was performed as described (de Almeida et al., 2017). To detect mitochondrial mass, cells were incubated with 50 nM MitoTracker Green (Fisher Scientific) at 37℃ for 25 min and processed according to manufacturer's protocol. A minimum of 20,000 cells were analyzed by flow cytometry using a BD LSRFortessa instrument and BD FACSDiva 6.2 software. Data were analyzed by FlowJo (v10). The median fluorescent intensities in each group were used to calculate the fold change compared to the CTRL group.

## Metabolic assays

The oxygen consumption rate (OCR) and extracellular acidification rate (ECAR) were performed by using the Seahorse Xfe96 Extracellular Flux analyzer (Agilent). Cells were plated in the Xfe96 cell culture plates at a density of 10,000 cells (HCC515) or 5000 cells (H2122) in 80 µl of growth medium overnight to generate a confluent monolayer of cells. The assay medium consisted of 10 mM glucose, 2 mM glutamine, 1 mM pyruvate in DMEM (D5030, MilliporeSigma). The reaction was monitored upon serial injections of oligomycin (5 µM), CCCP (1 µM), and antimycin A (2 µM). Results were normalized to total protein amount as determined by Pierce BCA Protein Assay Kit. Non-mitochondrial respiration is subtracted from the presented data. COX6B2-depleted cells were assessed following eight days of shRNA or three days of siRNA. Complex IV activity of isolated mitochondria was directly assessed in permeabilized cells as described (Salabei et al., 2014; Divakaruni et al., 2014), except that cells were permeabilized by XF PMP (1 nM). Complex III was inhibited with antimycin A (2 µM) and then treated with complex IV substrates (Ascorbate 10 mM, TMPD 100 µM) (Figure 3A).

## Measurement of ATP

Cell-Titer Glo (Promega) (CTG) was performed by manufacture's protocol but modified to use 15 µl of CTG for 100 µl of cells in media. Data were normalized to total protein amount as determined by Pierce BCA Protein Assay Kit. Luminescence was read using a Pherastar Plus plate reader (BMG Labtech).

## NAD$^+$/NADH assay

Cells were plated into 96-well tissue culture plates (Corning), allowed to adhere overnight and reach 80–90% confluency. Cells were processed according to the manufacturer's protocols for NAD$^+$/NADH-Glo Assay (Promega) with modification as previously described (Gui et al., 2016). NAD$^+$/NADH measured in COX6B2 depleted cells were transduced with shRNA for 8 days.

## Blue native PAGE (BN-PAGE)

Mitochondria isolation and BN-PAGE were performed as previously described (Jha et al., 2016; Wittig et al., 2006; Ikeda et al., 2013). The protocol was modified in that mitochondrial fractions were collected in buffer containing 200 mM sucrose, 10 mM Tris and 1 mM EGTA/Tris. Mitochondria (50 µg) were solubilized in 20 µl buffer containing 50 mM NaCl, 50 mM Imidazole/HCl, 2 mM 6-Aminocaproic acid, 1 mM EDTA, and digitonin (digitonin/protein ratio of 6 g/g) for 30 min at 4℃. Solubilized proteins were supplemented with Coomassie brilliant blue G-250 digitonin/Coomassie dye in a ratio of 4:1, separated on NativePAGE Bis-Tris gels (3–12%, Thermo Fisher Scientific), and stained by Colloidal Blue (Thermo Fisher Scientific) or immunoblotted. NativeMark Unstained Protein Standard (Thermo Fisher Scientific) was used as a marker (Mourier et al., 2014).

## Measurement of reactive oxygen species (ROS)

The Fluorescent probe MitoSOX Red(ThermoFisher) was used to measure superoxide. MitoSOX Red is a mitochondrial membrane-dependent dye and therefore could not be used in COX6B2-depleted

cells as they lose mitochondrial membrane potential ($\Delta\Psi_m$). In this case, Chloromethyldichlorodihydrofluorescein diacetate (CM-H2DCFDA, Fisher Scientific), detecting $H_2O_2$, was used for total ROS analysis in cells 11 days of shRNA silencing. For MitoSOX Red assay, $3 \times 10^5$ cells were stained by MitoSOX Red (5 µM; in HBSS) at 37°C for 10 min. As a positive control, cells were treated with 50 µM antimycin A during staining. For $H_2O_2$ measurements, cells reaching approximately 80% confluency in six-well dishes were stained by CM-H2DCFDA (5 µM; in HBSS) at 37°C for 20 min, washed with HBSS, and then recovered in culture media at 37°C for 15 min. Cells were immediately analyzed by flow cytometry using a BD LSRFortessa instrument and BD FACSDiva 6.2 software. A minimum of 20,000 cells were analyzed per condition. Data were analyzed by FlowJo (v10). Forward scatter (FSC) and side scatter (SSC) parameters were used to exclude cellular debris, dead cells, and doublets to retain viable single cell events. The median fluorescent intensities in each group were used to calculate the fold change compared to the CTRL group. CM-H2DCFDA had to be used in COX6B2-depleted cells because MitoSOX Red is dependent upon mitochondrial integrity, which is lost under these conditions (*Zielonka and Kalyanaraman, 2010*).

## EdU incorporation assays

Cells were exposed to ethynyl deoxyuridine (EdU) for 1 hr before fixing the cells in 4% formaldehyde. Cells were stained using the protocol for Click-iT EdU Alexa Fluor 488 Imaging Kit (Thermo Fisher) and co-stained with Hoechst 3342. Cells were quantified using fluorescence microscopy. EdU detected in COX6B2-depleted cells were transduced with shRNA for 9 days.

## Proliferation measurements

Cell proliferation was measured as described (*Gui et al., 2016*). HCC515 and H2122 cells were plated in replicates in 96-well plate with an initial seeding density of 1,200 cells and 700 cells, respectively. The proliferation rate was calculated using the following formula: Proliferation Rate (Doublings per day) = $\text{Log}_2$ (Final cell count (day 5)/Initial cell count (day 1))/4 (days).

## Soft-agar assays

Cells were seeded at a density of 50,000 cells (HCC515) or 5000 cells (H2122) per 12-well plate and treated as previously described (*Gallegos et al., 2019*). After 3 weeks, colonies were stained overnight with 0.01% crystal violet. Images were captured with a dissecting microscope, and quantitated by ImageJ software.

## Gene expression

RNA was isolated using a mammalian total RNA isolation kit (RTN350, Sigma) and treated with Dnase I (AMPD1, Sigma) according to the manufacturer's instructions. Equal microgram volumes of RNA were reverse transcribed for each experimental condition by the use of a High-Capacity cDNA reverse transfection kit (4368813, ThermoFisher Scientific) with oligo dT (12577–011, ThermoFisher Scientific). The resulting cDNA was used for expression analysis performed with an Applied Biosystems real-time PCR system with TaqMan real-time PCR probes (Thermo Fisher). The TaqMan qPCR probes were as follows: *COX6B2* (Hs00376070_m1), *COX6B1* (Hs01086739_g1). *RPL27* (Hs03044961_g1) was used as an internal loading assay for all expression assays.

## HIF-1 chromatin immunoprecipitation (ChIP) dataset analysis

Raw reads were obtained from GSE59937 (*Zhang et al., 2015*) and aligned to the human genome (hg19) using Bowtie2 with the 'sensitive' parameters enabled. SAMtools was used to sort and index mapped read files, which were then converted to bigWig files (DeepTools) for visualization on the IGV genome browser (*Robinson et al., 2011*).

## Transfections

siRNAs were reverse transfected with Lipofectamine RNAiMAX (13778, Thermo Fisher Scientific) following the manufacturer's instructions. siRNAs against *COX6B2* (sense sequence: COX6B2#1: 5'-GGAACGAGCAGATCAAGAA-3'; COX6B2#2: 5'-GCCAGAACCAGATCCGTA A-3') were purchased from Dharmacon and Sigma respectively. siGENOME Non-Targeting siRNA Pool #2 was used as a negative control (siCTRL). sgRNAs were forward transfected in six-well dishes by using 6 µl of

FuGENE HD (PRE2311, Fisher Scientific) with 3 µg of sgRNA. Guide sequences of COX6B2 sgRNA are as following:

> oligo #1: 5'-CACCGCGGCGTCGACCATTTCCCCTTGG-3',
> oligo #2: 5'-CACCGCGTCGACCATTTCCCCTTGGGG-3',
> oligo #3: 5'-CACCGACATCCAACATCCACGAAGGAGG-3'.

## Measurement of mitochondrial membrane potential

Following 6 days of shRNA ($3 \times 10^5$ cells), cells were collected and resuspended in HBSS containing 0.2 µM JC-1 and incubated at 37°C in a $CO_2$ incubator for 25 min. The stained cells were collected by centrifuge and washed twice with HBSS containing 1% FBS. Then cells were resuspended in 500 µl of 1% FBS HBSS buffer. Cells were immediately analyzed by flow cytometry using a BD LSRFortessa instrument and BD FACSDiva 6.2 software. A minimum of 20,000 cells were analyzed per condition. FlowJo (v10) was used to generate scatter plots.

## Annexin-V staining

After 7 days of shRNA-mediated knockdown $1 \times 10^5$ cells were wash once in PBS and resuspend in Annexin-V Binding Buffer (422201, BioLegend) with 3 µl of Annexin-V-APC (640941, BioLegend) and DAPI (200 ng/ml), and incubated for 15 min at room temperature in the dark. Cells were immediately analyzed by flow cytometry using a BD LSRFortessa instrument and BD FACSDiva 6.2 software. A minimum of 20,000 cells were analyzed per condition. FlowJo (v10) was used to generate scatter plots. Early-stage apoptotic cells are presented in the lower right quadrants (Annexin-V positive and DAPI negative) and late-stage apoptotic cells are presented in the upper right quadrants (Annexin-V positive and DAPI positive).

## Senescence-associated β-galactosidase activity

Cells were stained by using Senescence Beta-Galactosidase Staining Kit following 11 days of shRNA (9860, Cell Signaling Technology) according to the manufacture's protocol. Cultures were examined under phase-contrast microscopy to determine the percentage of positive cells with blue precipitate.

## Xenograft experiments

All animal experiments were conducted under a UT-Southwestern IACUC approved protocol. Hsd: Athymic Nude-Foxn1[nu] female mice were purchased from Envigo. At six to eight-weeks of age (23–25 grams) mice were randomly assigned to the indicated groups. Sample size was determined by the formula: $n = 1+2C(s/d)^2$, in which C = 10.51, based on a power of 90% and a confidence interval of 95%, S = standard deviation of tumor volume (20%), D = tumor burden difference to be considered significant (50%). Therefore, five mice per condition were tested. $2 \times 10^6$ cells in 100 µl PBS were subcutaneously injected in the flank. Once tumors were visible, tumor volume was measured by caliper in two dimensions, and volumes were estimated using the equation V = length x width$^2$/2. Caliper measurements were made twice a week.

## Statistical analysis

Graphpad Prism (Graphpad Software) was used to perform all statistical analyses. Data were assessed for normality by Shapiro-Wilk tests. Unpaired Student's $t$-test was used for data with a normal distribution. Mann-Whitney test was used for data that did not conform to a normal distribution. p-Values less than or equal to 0.050 were considered significant. Biological replicates represent experiments performed on samples from separate biological preparations; technical replicates represent samples from the same biological preparation run in parallel.

## Acknowledgements

The authors thank Melanie Cobb for critical review of the manuscript and Jonathan Friedman for technical suggestions. AWW, CC, ZG and KM were supported by NIH (R01CA196905). AWW and JW were supported by SU2C (SU2C-AACR-IRG1211). PM was supported by The Cancer Prevention and Research Institute of Texas (CPRIT) RP180778. Human tissue samples were obtained and stained

with the help of the UTSW tissue management shared resource and the UTSW Biomarker Research Core ,which are supported by the Simmons Cancer Center Core grant from National Cancer Institute (P30CA142543).

# Additional information

### Competing interests

Joshua Wooten: Joshua Wooten is affiliated with Nuventra. The author has no financial interests to declare. The other authors declare that no competing interests exist.

### Funding

| Funder | Grant reference number | Author |
|---|---|---|
| National Cancer Institute | R01CA196905 | Chun-Chun Cheng<br>Kathleen McGlynn<br>Angelique W Whitehurst |
| National Cancer Institute | P30CA142543 | Chun-Chun Cheng<br>Joshua Wooten<br>Kathleen McGlynn<br>Angelique W Whitehurst |
| Cancer Prevention and Research Institute of Texas | RP180778 | Prashant Mishra |

The funders had no role in study design, data collection and interpretation, or the decision to submit the work for publication.

### Author contributions

Chun-Chun Cheng, Conceptualization, Formal analysis, Validation, Investigation, Methodology, Writing - original draft, Writing - review and editing; Joshua Wooten, Formal analysis, Investigation, Methodology; Zane A Gibbs, Investigation; Kathleen McGlynn, Data curation, Validation, Investigation; Prashant Mishra, Conceptualization, Formal analysis, Supervision, Investigation, Writing - review and editing; Angelique W Whitehurst, Conceptualization, Supervision, Funding acquisition, Investigation, Visualization, Methodology, Writing - original draft, Writing - review and editing

### Author ORCIDs

Chun-Chun Cheng  https://orcid.org/0000-0002-3657-8715
Zane A Gibbs  http://orcid.org/0000-0003-0294-1878
Angelique W Whitehurst  https://orcid.org/0000-0002-9505-0240

### Ethics

Animal experimentation: This study was performed in strict accordance with the recommendations in the Guide for the Care and Use of Laboratory Animals of the National Institutes of Health. All of the animals were handled according to approved institutional animal care and use committee (IACUC) protocols (2016-101795) at UTSW.

### Decision letter and Author response

Decision letter https://doi.org/10.7554/eLife.58108.sa1
Author response https://doi.org/10.7554/eLife.58108.sa2

# Additional files

### Supplementary files

• Transparent reporting form

### Data availability

All data generated or analyzed during this study are included in the manuscript and supporting files. Sequencing data used was previously published by another group and is referenced.

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
