## [Decision Letter]

**Acceptance summary:**

This work is significant in identifying COX6B2, a subunit of complex IV of the electron chain normally only expressed in testis, as being upregulated in lung cancer and contributing to increased tumor growth and poor patient outcomes in lung cancer. This work is an example of cancers hijacking mechanisms usually restricted to a developmental or tissue-specific context in order to promote tumor growth.

**Decision letter after peer review:**

Thank you for submitting your article "Sperm-specific COX6B2 enhances oxidative phosphorylation, proliferation, and survival in lung adenocarcinoma" for consideration by *eLife*. Your article has been reviewed by three peer reviewers, one of whom is a member of our Board of Reviewing Editors, and the evaluation has been overseen by Maureen Murphy as the Senior Editor. The following individual involved in review of your submission has agreed to reveal their identity: Alexander Muir (Reviewer #2).

The reviewers have discussed the reviews with one another and the Reviewing Editor has drafted this decision to help you prepare a revised submission.

Summary:

This work examines how lung cancer cells co-opt a sperm-specific isoform of COX6B, namely COX6B2, to promote OXPHOS, energy production and growth under nutrient poor conditions. The authors over-express or knockdown expression of COX6B2 in 2 different human lung adenocarcinoma cells lines and provide evidence that they claim shows that COX6B2 promotes tumor cell growth by promoting electron transport supercomplex formation, NAD+ and ATP production, limits ROS and promotes proliferation in vitro and tumor growth in SQ xenografts. Significantly, they suggest that COX6B2 is induced by hypoxia and allows tumor cells to survive in those conditions.

Essential revisions:

The reviewers all agreed that the following essential revisions would be required before the manuscript would be acceptable:

1) The data indicating that COX6B2 is upregulated in lung cancers was not found to be convincing and the authors are requested as a minimum to use publicly available databases (such as TCGA and CCLE) to enhance this claim by comparing COX6B2 levels in normal and tumor tissue. The addition of IHC staining for COX6B2 levels on normal and tumor tissue would also strengthen the message and is also requested.

2) Data indicating that COX6B2 promotes ETC supercomplex formation was not found to be convincing, and additional experiments including imaging (IF for ETC components and/or EM for cristae density) and biochemical assays (measurement of Km and Kcat) on isolated mitochondria/cells to examine how COX6B2 versus COX6B1 affects activity of complex IV is required.

3) Greater analysis of the inter-relationship between COX6B1 and COX6B2 is needed for clarity, including an explanation of how they are related to each other and what is currently know about their regulation – how does the expression of one change when the other is knocked down or over-expressed. In particular, analysis of both COX6B1 and COX6B2 protein in the genetically modified lines versus controls in regular versus hypoxic conditions would be required to address why cells are not able to survive hypoxic stress despite apparently inducing endogenous COX6B2 while COXB2 over-expressing lines are able to survive.

4) There is overall some concern with the cell lines used with only one clone of each line using one set of shRNAs. These experiments do not control properly for off-target effects of shRNAs and there is some concern given the small scale of some effects that what is observed is due to those effects and not loss of COX6B2. Thus, it will be important for the authors repeat key experiments with either KO or knockdown lines that are reconstituted with COX6B2 to determine whether it can revert the phenotypes observed by knockdown with shRNAs. For example, the authors could use lines in which COX6B2 is CRISPRed out and then reconstituted with wild-type COX6B2 (or mutant COX6B2?), or alternatively, with lines utilizing multiple different sets of shRNA (targeting different defined regions of the mRNA) and again reconstituting with COX6B2.

---

## [Author Response]

Essential revisions:The reviewers all agreed that the following essential revisions would be required before the manuscript would be acceptable:1) The data indicating that COX6B2 is upregulated in lung cancers was not found to be convincing and the authors are requested as a minimum to use publicly available databases (such as TCGA and CCLE) to enhance this claim by comparing COX6B2 levels in normal and tumor tissue. The addition of IHC staining for COX6B2 levels on normal and tumor tissue would also strengthen the message and is also requested.

As requested, we have included mRNA data from TCGA comparing COX6B2 expression in normal and lung adenocarcinoma tissues. This analysis shows a statistically significant (< 0.0001) upregulation of COX6B2 in tumor tissues (Figure 1D). We have also included COX6B2 IHC staining of 22 normal and 32 tumor tissues (Figure 1G). Here, we were unable to detect COX6B2 expression in normal tissues, but observed robust expression in all but one of the tumor tissues. This data is consistent with the immunoblots presented in Figure 1E from the original submission.

2) Data indicating that COX6B2 promotes ETC supercomplex formation was not found to be convincing, and additional experiments including imaging (IF for ETC components and/or EM for cristae density) and biochemical assays (measurement of Km and Kcat) on isolated mitochondria/cells to examine how COX6B2 versus COX6B1 affects activity of complex IV is required.

We agree that activity measurements of complex IV are important to the study and have included these for complex IV isolated from COX6B2 or COX6B1 overexpressing LUAD cells (Figure 3B-C). Here, we find that COX6B2 overexpression confers a significant increase (~25%) in complex IV activity. In contrast, a minimal, statistically insignificant increase in activity was observed following COX6B1 overexpression.

To more comprehensively examine supercomplex formation, we quantitated monomer, dimer and supercomplexes containing either endogenous and overexpressed COX6B1 or COX6B2 over multiple, independent experiments (Figure 3D-G). Collectively, this analysis demonstrates that ~90 % of COX6B2 (endogenous or overexpressed) is found in complex IV dimers or supercomplexes. This is in stark contrast to COX6B1, of which the majority of either endogenous or overexpressed is present in monomeric complex IV, even though there is a redistribution following overexpression. The data indicate that COX6B2 (but not COX6B1) preferentially incorporates into supercomplexes.

To further evaluate whether supercomplex formation was changing upon COX6B2 expression, we measured the distribution of the complex IV subunit, COXIV, in these complexes following COX6B1-V5 or COX6B2-V5 overexpression. Here, we found that ~40% of COXIV was in supercomplexes in the COX6B2-V5 expressing cells, while only 18% was present in supercomplexes in COX6B1-V5 expressing cells. These complex increases likely reflect the concomitant increase in OCR that we observe in COX6B2 overexpressing, but not COX6B1 cells. Together these results indicate that 1) COX6B2 preferentially integrates into dimeric complex IV and supercomplexes and 2) the expression of COX6B2 appears to shift complex IV to higher order complexes as compared to COX6B1. We have included immunofluorescence of complex IV that indicates no change in complex IV expression as we would expect (Figure 3—figure supplement 1A). The text has been revised to more precisely describe these observations.

3) Greater analysis of the inter-relationship between COX6B1 and COX6B2 is needed for clarity, including an explanation of how they are related to each other and what is currently know about their regulation – how does the expression of one change when the other is knocked down or over-expressed. In particular, analysis of both COX6B1 and COX6B2 protein in the genetically modified lines versus controls in regular versus hypoxic conditions would be required to address why cells are not able to survive hypoxic stress despite apparently inducing endogenous COX6B2 while COXB2 over-expressing lines are able to survive.

We have included a statement indicating that transcriptional regulation of COX6B1 and B2 has not previously been studied in the Introduction. Per the reviewer’s request, we measured COX6B1 and COX6B2 protein expression when the corresponding isoform is overexpressed or knocked down and did not observe any significant changes (Figure 1—figure supplement 1B). Furthermore, we have also included a measurement of COX6B1 and B2 protein accumulation in hypoxia following overexpression of the corresponding isoform (Figure 4—figure supplement 1E). The data indicate that COX6B1 expression is not affected by COX6B2-V5 overexpression in normoxia or hypoxia and vice-versa.

With regard to the reviewer’s comment that the control cells are not able to survive hypoxic stress despite inducing endogenous COX6B2, we point out that this experiment measures proliferation (by EdU) not survival. We have now specified this point in the text. The control cells, inducing endogenous COX6B2, do continue to proliferate, albeit at ~40% of the original rate (this was incorrectly noted as a 30% decrease in the original text and has been corrected).

4) There is overall some concern with the cell lines used with only one clone of each line using one set of shRNAs. These experiments do not control properly for off-target effects of shRNAs and there is some concern given the small scale of some effects that what is observed is due to those effects and not loss of COX6B2.

We think there was some confusion here. We performed loss of function experiments by three independent mechanisms: 1) transient transfection of siRNAs (2 independent siRNA sequences alone (Figure 5D) and in a pool (Figure 5B)); 2) lentiviral introduction of shRNAs (four independent shRNAs in a pool) and 3) transient transfection of sgRNA to perform CRISPR. The siRNA, shRNA and sgRNA sequences are all distinct from one another and target different parts of the COX6B2 mRNA. As stated in the manuscript, we were unable to establish a clone for stable knockdown (shRNA) or knockout (sgRNA) largely due to the loss of viability of these populations over time. We now indicate the method of knockdown as “siRNA”, “shRNA”, or “sgRNA” throughout the figures and text. We indicate siRNA or shRNA in the figure and legends for Figures 5-7. The sequences are all listed in the Materials and methods section. To help dispel confusion, we have noted “pool” for siRNA and shRNA where these pools were used in Figures 5-7. In addition, we have clarified this approach in the text that refers to these figures.

Thus, it will be important for the authors repeat key experiments with either KO or knockdown lines that are reconstituted with COX6B2 to determine whether it can revert the phenotypes observed by knockdown with shRNAs. For example, the authors could use lines in which COX6B2 is CRISPRed out and then reconstituted with wild-type COX6B2 (or mutant COX6B2?), or alternatively, with lines utilizing multiple different sets of shRNA (targeting different defined regions of the mRNA) and again reconstituting with COX6B2.

We appreciate the reviewer’s broader point regarding off-target effects and that the overexpression experiment suggested is considered a gold standard. However, our gain of function experiments indicate that increasing COX6B2 expression leads to enhanced oxidative phosphorylation and proliferation (Figures 2 and 4). Given these gain of function phenotypes, introduction of COX6B2 in knockdown cells would likely work irrespective of off-target effects of the silencing mechanism. This issue was why we originally used siRNA, shRNA and sgRNA (CRISPR) to cross validate the loss of function phenotypes through multiple, distinct approaches. To the reviewer’s point, we have performed siRNA in the COX6B2-V5 overexpressing cells. Here, the increase in COX6B2 expression (2-fold) scales with a rescue of cleaved caspase-3 (50%), thus providing an indication that the siRNA phenotype of cleaved caspase-3 can be rescued by increasing COX6B2 expression. This figure is now in Figure 6—figure supplement 1C. To summarize, we have used three independent knockdown methods, each phenocopying one another. In addition, we use gain of function approaches, which exhibit the converse phenotype when COX6B2 but not when COX6B1 is overexpressed.